# Metabolic Crossroads: Unveiling the Complex Interactions between Obstructive Sleep Apnoea and Metabolic Syndrome

**DOI:** 10.3390/ijms25063243

**Published:** 2024-03-13

**Authors:** Aisling Heffernan, Darko Duplancic, Marko Kumric, Tina Ticinovic Kurir, Josko Bozic

**Affiliations:** 1Department of Pathophysiology, University of Split School of Medicine, 21000 Split, Croatia; aisling-heffernan@hotmail.com (A.H.); marko.kumric@mefst.hr (M.K.); tticinov@mefst.hr (T.T.K.); 2Department of Cardiology, University Hospital of Split, 21000 Split, Croatia; darko.duplancic@mefst.hr; 3Laboratory for Cardiometabolic Research, University of Split School of Medicine, 21000 Split, Croatia; 4Department of Endocrinology, Diabetes and Metabolic Disorders, University Hospital of Split, 21000 Split, Croatia

**Keywords:** obstructive sleep apnoea, metabolic syndrome, diabetes mellitus, obesity, leptin

## Abstract

Obstructive sleep apnoea (OSA) and components of metabolic syndrome (MetS) are inextricably connected. Considering the increasing burden of MetS and OSA, in the present review, we aimed to collate and summarise the potential pathophysiological mechanisms linking these pathologies. In short, obesity appears to promote OSA development via multiple pathways, some of which are not directly related to mass but rather to metabolic complications of obesity. Simultaneously, OSA promotes weight gain through central mechanisms. On the other hand, diabetes mellitus contributes to OSA pathophysiology mainly through effects on peripheral nerves and carotid body desensitization, while intermittent hypoxia and sleep fragmentation are the principal culprits in OSA-mediated diabetes. Apart from a bidirectional pathophysiological relationship, obesity and diabetes mellitus together additively increase cardiovascular risk in OSA patients. Additionally, the emergence of new drugs targeting obesity and unequivocal results of the available studies underscore the need for further exploration of the mechanisms linking MetS and OSA, all with the aim of improving outcomes in these patients.

## 1. Introduction

Obstructive sleep apnoea (OSA) is an increasingly common sleep-related breathing disorder. The condition is characterised by repetitive collapse of the pharyngeal airway during sleep, precipitating episodes of hypopnea or apnoea, oxyhaemoglobin desaturation and sleep fragmentation [1]. In clinical practice and research, OSA severity is most often quantified by the apnoea–hypopnea index (AHI), i.e., the mean number of hypopneas and apnoeas per hour [2]. In adults, mild, moderate and severe OSA correlates to 5–15 events/h, 15–30 events/h and ≥30 events/h, respectively [3]. Despite increasing recognition of OSA and the myriad of conditions of negative health sequalae, there is a paucity of literature studies pertaining to global prevalence [4]. In the adult population, OSA has been estimated to afflict ~936 million individuals aged 30–69; however, considerable variation exists depending on the diagnostic criteria employed [4,5]. 

Metabolic syndrome (MetS), once dubbed by researchers “the deadly quartet”, comprises a constellation of metabolic and cardiovascular abnormalities, including central adiposity, insulin resistance, hypertension and dyslipidaemia [6,7]. Rather than a distinct disease entity, MetS represents a series of risk factors which predispose individuals to cardiometabolic disease. The assessment of six clinical parameters, i.e., (1) waist circumference, (2) fasting glucose levels, (3) triglyceride levels, (4) high-density lipoprotein levels, (5) cholesterol and (6) blood pressure, enables the diagnosis of MetS [8]. Similar to those of OSA, the consensus definitions and diagnostic criteria of MetS have evolved over time, making accurate prevalence estimates difficult [9]. Globally, MetS prevalence ranges from <10 to 80% of the population, depending on region, population characteristics and the diagnostic criteria utilised [10]. 

The frequent co-existence of MetS and OSA previously led to the description of “Syndrome Z”, though this has now been largely abandoned [11]. Sharing a similar pathophysiological milieu, a bi-directional relationship between the two has been proposed, with each condition being suggested to independently influence and aggravate the development and clinical consequences of the other [11]. Obesity is a common aetiological factor for MetS and OSA, and the prevalence of both conditions tends to increase in concordance with the global epidemic of obesity [12]. 

Left untreated, MetS and OSA are significantly associated with cardiovascular disease (CVD), which remains the leading cause of morbidity and mortality worldwide [12,13]. Given the rising incidence of MetS and OSA, alongside the global burden of CVD, it is imperative to explore the pathophysiological cross-talk between these pathologies in an effort to ameliorate future CVD risk. As such, the purpose of this narrative review is to collate and summarise the potential pathophysiological mechanisms linking OSA and MetS which may have important implications for future treatment strategies.

## 2. Obesity, OSA and MetS: Cause or Consequence?

Obesity is defined as a body mass index (BMI) ≥ 30 kg/m^2^ [14]. Several studies have demonstrated the role of obesity in the pathogenesis of OSA; at least half of adult OSA is directly ascribable to excess weight [3]. In patients with a BMI greater than 28 kg/m^2^, OSA has a prevalence of 41%, and among individuals referred for bariatric surgery, it exceeds 78% [3,15,16]. While it is widely accepted that obesity is a potent risk factor for OSA, waist circumference, a component of metabolic syndrome and proxy of central adiposity, and neck circumference (NC) may be of greater predictive and prognostic value in OSA [17]. A recent study illustrated that in subjects with OSA and visceral fat accumulation (VFA), visceral fat measured using abdominal bioelectrical impedance analysis (A-BIA) was a more significant correlate of OSA severity than BMI [18]. Similarly, a cross-sectional analysis of 120 obese individuals subject to biochemical and anthropometric assessment alongside polysomnography demonstrated that NC is independently associated with both MetS and OSA. Notably, an NC ≥ 38 cm had sensitivity of 54% and 58% and specificity of 70% and 79% in predicting the presence of MetS and OSA, respectively [19]. In clinical terms, although the presence of normal NC does not exclude OSA, the presence of large NC constitutes a significant risk factor for OSA. Moreover, visceral fat accounts for up to 10–20% of total adiposity in men and 5–8% in women. This propensity for a more “central” deposition of fat in males may explain the sex dimorphism in OSA prevalence [20]. As such, it is likely that OSA is more aptly described as a manifestation of visceral and neck adiposity than total adiposity (visceral fat and subcutaneous fat), which BMI describes [3]. A combination of local mechanical factors affecting upper airway (UA) patency, described as “direct weight-dependant mechanisms”, and the metabolic activity of visceral abdominal fat (VAF) appear to likely explain these observations [21].

The physical impact of excess adiposity can alter normal upper airway mechanics and contribute to OSA pathogenesis through various mechanisms [16]. Central adiposity is associated with parapharyngeal fat deposition and enlargement of surrounding soft tissues, which create a relative stenosis of the upper airway (UA), thereby increasing upper airway (UA) collapsibility and predisposing to apnoea [22]. UA collapsibility is accepted as the primary pathogenic mechanism in OSA, and susceptibility to collapse can be assessed by pharyngeal critical closing pressure (Pcrit), which is higher in obese patients [23,24,25,26]. Moreover, it is likely that the reduced lung volumes induced by central adiposity further impair UA patency in addition to compromising gas exchange [22,27]. However, the aforementioned “direct weight-dependant” mechanisms offer only a partial explanation of OSA pathogenesis, as the condition also affects lean individuals, and likewise not all obese individuals have OSA [28].

Interestingly, evidence indicates that the relationship between obesity and OSA is reciprocal, and that the presence of OSA may itself promote weight gain and obesity [29]. While the precise mechanisms underlying this bidirectional relationship remain to be fully elucidated, various hypotheses have been suggested. Energy balance (EB) is complex and multifaceted, involving food intake, the hormonal regulation of appetite/hunger/satiety and energy expenditure via metabolism and physical activity [30]. To incite weight gain, energy intake (EI) must exceed energy expenditure (EE) [30]. In the Sleep Heart Health Study (SHHS), OSA severity correlated with a 5-year increase in BMI after correcting for known confounders [28]. Furthermore, in a weight loss intervention study, participants with a greater AHI (≥5) lost 2.2 ± 0.9% less weight than those with a lower AHI (<5) [31]. These findings are at odds with studies demonstrating that resting metabolic rate (RMR), which represents the largest component of EE, is paradoxically increased in patients with OSA secondary to nocturnal arousal, sympathetic overdrive and increased effort in breathing, which in theory could preclude weight gain [31,32]. Evidence that continuous positive airway pressure (CPAP), the primary treatment modality in OSA, results in reduced RMR lends further credence to these findings [32]. 

Nevertheless, it appears that the heightened EE observed in OSA is largely offset by various other OSA-induced EB alterations [30]. First, sleep deprivation associated with OSA results in physical inactivity, and a negative association between OSA severity and physical activity level exists [33]. Second, ghrelin, which has orexigenic action (i.e., stimulates food intake), is abnormally elevated in OSA, and the satiety signalling function of leptin is abrogated in OSA, consistently with a hormonal profile predisposing to high EI [34]. Additionally, beyond the hormonal regulation of hunger, emerging evidence posits that reduced sleep time engenders alterations in the brain’s processing of food cues. More specifically, when assessed by functional MRI, food cues invoked heightened activity in brain regions related to reward and pleasure seeking in periods of restricted sleep when compared with habitual sleep, which, in one study, led to subsequent increases in energy intake [35,36]. Further research is required to enhance our comprehension of how sleep restriction impacts the brain’s neural processing of food and whether these processes affect future food choice.

The role of the aforementioned “direct weight-dependant mechanisms” in OSA aetiology is consequent to the physical or mechanical weight of excess adiposity. In addition, however, various “indirect weight-dependant mechanisms” have been proposed, the most studied of which are glucose intolerance, insulin and leptin resistance (Figure 1) [21,28]. These metabolic aberrations are pathophysiological components of obesity or diabetes and may pose as both a risk factor for and a product of OSA [21,28]. It is well established that weight loss can ameliorate obesity-associated metabolic derangements and AHI severity [37]. However, this adds further complexity in discerning the relative contributions of direct and indirect weight-dependant mechanisms to OSA pathogenesis, as weight loss thus not only represents a change in the mechanical contribution of obesity but also the concomitant obese physiology.

## 3. OSA, Hyperglycaemia and Diabetes Mellitus

Impaired glycaemic control is a notable feature of obese physiology, and some studies have purported a two-way relationship and reverse causality between OSA and hyperglycaemia/diabetes [38]. Excess body weight is a common etiological factor in OSA and type 2 diabetes mellitus (T2DM) [39]. Therefore, whether the negative association of T2DM with OSA, and vice versa, is independent of adiposity or arises as a consequence of it remains unclear [5,40]. An unexpectedly high prevalence of OSA (40%) in non-obese type 1 diabetic individuals, not dissimilar to that of type 2 diabetic individuals (23–50%), however, implies that hyperglycaemia and glucose dysmetabolism may increase susceptibility to or accelerate progression of OSA [5,41]. In a 4-month-long small (n = 24) interventional study investigating whether glycaemic control optimisation therapy in type 2 diabetic individuals impacted sleep breathing parameters, Guiterrez-Carrasguilla et al. demonstrated that participants with a HbA1c reduction ≥0.5% had significant improvements in the AHI (26.1 (8.6 to 95.0) events/h at baseline vs. 20.0 (4.0 to 62.4) events/h at the end) [42]. Crucially, these findings were achieved without significant changes in BMI, WC and NC, and subjects exhibiting weight loss (reduction in BMI ≥ 2.0 kg/m^2^) were excluded to minimise potential confounding [42]. The above work expands on previous data from the same group which illustrated reversibility in the number of nocturnal oxygen desaturation events in type 2 diabetic individuals with five days of intensified glycaemic control [43]. Though the small sample sizes and short duration of both aforementioned studies limit their clinical applicability, they provide evidence that tightened glycaemic control can potentiate respiratory function. This may have important clinical implications for poorly controlled DM patients and may represent a potential therapeutic strategy to ameliorate OSA severity in T2D, which merits further larger-scale research. 

Various mechanisms have been put forward to explain the biologic plausibility of hyperglycaemia, independently of obesity, amplifying the risk of OSA. The prevailing literature implies that these effects may be mediated primarily through (1) development of neuropathy and (2) carotid body (CB) desensitisation secondary to chronic hyperglycaemia [5]. Neuropathy is a common sequalae of diabetes, estimated to afflict up to 50% of diabetic patients. Various patterns of diabetic neuropathy can present, including distal symmetric polyneuropathy (DSP), mononeuropathy and autonomic neuropathy (AN) [44]. There is elevated prevalence of OSA in diabetic individuals with AN compared with those without, with an estimated frequency of 26–30% in this cohort [45,46]. In addition, Bottini et al. demonstrated that obese diabetic individuals with AN have a more severe OSA phenotype (AHI of 39.5 ± 13) than obese diabetic individuals without AN (AHI of 15.8 ± 12) and age-matched obese controls (19.3 ± 21) [47]. These findings are consistent with other data showing that diabetic individuals with AN have prolonged apnoeic events and more profound nocturnal oxygen desaturation events [47,48,49]. It is possible that autonomic neuropathy impacts the chemical control of respiration by altering central and peripheral chemoreceptor signals, in addition to vagal, glossopharyngeal and proprioceptive nerves. Alternatively, UA neuropathy may evoke dysfunction of the UA dilator muscle, resulting in UA narrowing or closure, thereby predisposing to apnoea [5,50].

The carotid body (CB) is a bilateral sensory organ located at the bifurcation of the common carotid artery [51]. The CB governs the peripheral chemoreflex response and plays an imperative role in the chemical control of breathing. The ventilatory response to hypoxia (VHR) is initiated by the CB, which senses low arterial O_2_ and subsequently evokes depolarisation of glomus (type 1) cells and neurotransmitter release. The respiratory network of the brainstem receives afferent inputs from the glomus cells via the carotid sinus nerve, which projects to the nucleus solitarius and respiratory motor neurons to increase ventilation [51,52]. Rather than solely sensing pH and hypoxia as previously considered, emerging evidence indicates that the CB acts as a multipurpose sensor, responsive to several hormonal and metabolic stimuli, including leptin, insulin and glucose [52,53]. The capacity of the CB to directly sense glucose lacks consensus, with conflicting evidence that is drawn primarily from animal and in vitro studies [52,53,54,55]. Bin-Jaliah et al. found that hypoglycaemia in rats stimulated ventilation and increased the hypercapnic ventilatory response, effects which were subsequently eliminated by resection of the carotid sinus. However, when investigated in vitro with low levels of glucose in the superfusate, the firing rate of the carotid sinus nerve remained unchanged [54,55]. The authors concluded that rather than acting directly on the CB, glucose elicits an indirect effect mediated by other metabolic blood-borne factors [53,54,55]. Interestingly, a more recent human study sought to characterise the effect of hyperglycaemia on several physiologic parameters under CB regulation. Hyperglycaemia was induced in the study’s 13 participants, and variations in heart rate, blood pressure and ventilation under both normoxic and hypoxic conditions were investigated. The authors highlighted that the hyperglycaemic state attenuated responsiveness of the CB to hypoxia, with respiratory rates decreasing when compared with fasting euglycemic levels [56]. 

Although neuropathy and altered responsiveness of the CB represent plausible mechanisms linking hyperglycaemia to OSA aetiology, they remain largely speculative based on small-scale human studies, animal studies and in vitro research. With that said, recent observational data from the UK corroborate the supposition that dysglycaemia/T2D can predict incident OSA, independently of the BMI [57]. This association contrasts with other data which failed to identify any temporal relationship between T2D and incident OSA in a Taiwanese cohort [58]. Identifying predictors of OSA is crucial to aiding screening and ameliorating disease burden; therefore, adequately powered prospective studies are required to delineate the discrepancies existing in the current literature. 

On the other hand, several early reports indicated that impaired glucose tolerance and DM were epiphenomena of OSA rather than etiological factors [59]. The majority of these preliminary studies were, however, encumbered by methodological limitations, with subjects self-reporting symptoms to characterise OSA, inadequate adjustment of key confounding variables such as obesity and small sample sizes [48,59]. Subsequent observational studies overcame these limitations with the use of polysomnography for OSA diagnosis and by statistically accounting for BMI and/or WC to investigate the contribution of obesity [48]. Currently, various lines of evidence position OSA as a precursor of insulin resistance, glucose intolerance and T2DM, independently of obesity [60]. In cross-sectional studies, these metabolic abnormalities correlate with AHI severity, though the degree of nocturnal oxygen desaturation appears to be the main determining factor [60]. The downstream pathways of two cardinal features of OSA, namely, intermittent hypoxia (IH) and sleep fragmentation (SF), likely underlie the associations of OSA with abnormal glucose homeostasis [40]. Accordingly, studies in rodents modelling the cyclical bouts of deoxygenation/reoxygenation which occur in OSA demonstrate that IH disrupts glucose homeostasis [61]. Likewise, male and female study participants exposed to IH show altered insulin sensitivity and glucose intolerance [62,63]. 

Aside from IH, the interruption of sleep continuity, or SF, emanating from recurrent arousals in OSA negatively impact insulin sensitivity, though empirical data on these effects of SF are somewhat scant [48]. The downstream pathways of IH and SF which incite glucose dysmetabolism are not yet well clarified; however, potential mediators may include sympathetic hyperactivity, oxidative stress and concomitant inflammation, hypothalamic–pituitary–adrenal axis (HPA) dysfunction, IGF-1 and the role of adipokines [3,5,40,48]. Reflective of the raised sympathetic activity from the combined effects of IH and SF, OSA patients have elevated plasma and urinary catecholamines, which can be effectively reduced by CPAP therapy [64]. Catecholamines increase plasma glucose by activation of the β-adrenergic receptor in the liver, which augments glycogenolysis via cAMP activation and gluconeogenesis indirectly by increasing substrate availability [65]. At the same time, insulin-mediated glycogenesis is inhibited [65,66]. Aside from creating new glucose and mobilising endogenous stores for blood-stream entry, catecholamines also reduce insulin-mediated glucose uptake in tissues other than the CNS [67]. Moreover, sympathetic activity exerts lipolytic effects, increasing the levels of circulating plasma free fatty acids, which further hinders insulin sensitivity [66]. In these ways, the upregulation of the SNS in OSA may participate in patient’s trajectory toward IR and DM. Oxidative stress, the phenomenon characterised by an altered balance between production of reactive oxygen species (ROS)/free radicals and the body’s anti-oxidant defence systems is well documented in OSA, and elevated OS biomarkers, including 8-isoprostane and 8-Hydroxy-2-deoxyguanosine, have been observed in patients [68,69]. Nightly episodes of IH followed by reoxygenation in OSA establish the oxidative imbalance, analogous to that of ischemia/reperfusion injury, generating ROS and activating the inflammatory cascade [70]. The oxidative milieu and free radicals are thought to play a central role in IR and DM, exerting deleterious effects via B-cell dysfunction/apoptosis, decreased GLUT-4 expression, mitochondrial dysfunction, increased inflammation and altered insulin signalling pathways [71]. Thus, OS represents a potentially important mechanism involved in OSA-impaired glucose tolerance. Previous reports suggest that recurrent arousals in OSA activate the HPA axis and increase the levels of the primary human glucocorticoid—cortisol [72]. Cortisol regulates several aspects of glucose homeostasis by inhibiting insulin release, antagonising insulin-mediated glucose uptake/utilisation and promoting gluconeogenesis in the liver [73]. Thus, excess cortisol causes IR and hyperglycaemia [73]. However, available studies investigating the relationship between cortisol levels and OSA have produced mostly inconsistent findings [72,74]. Moreover, many of this data are burdened by methodological concerns, including the single-timepoint sampling of cortisol or the inconsistent timing of sample collection, thus neglecting the innate circadian variation in cortisol [74]. In addition, existing studies often failed to adequately match OSA patients to control groups based on covariates recognised to alter cortisol profiles such as age and obesity [75,76]. In future research, repeated cortisol sampling, greater attention to cofounders such as age and body composition and the use of challenge tests to establish HPA axis activation may enable more definitive conclusions to be drawn on the role of HPA physiology in OSA-related metabolic dysfunction [74]. 

Interestingly, in the last decade, studies have revealed an association between subnormal insulin-like growth factor-1 (IGF-1) levels and various MetS components, including obesity, dyslipidaemia, IR and hyperglycaemia [77,78]. Under physiological conditions, growth hormone (GH) is the principal regulator of hepatic IGF-1 secretion, which mimics the action of insulin. Reduced IGF-1 levels in OSA have been observed to decrease peripheral glucose uptake and stimulate hepatic gluconeogenesis, in turn promoting IR [77]. MetS is highly prevalent among patients with adult-onset GH deficiency, and studies indicate that IGF-1 levels are inversely associated with MetS in OSA patients secondary to hypoxemia [79,80]. Furthermore, low IGF-1 levels are associated with reduced adiponectin—an adipocyte-derived factor with insulin-sensitising properties, which is also decreased in OSA patients [77,81]. Although large-scale research is currently lacking, in the future, IGF-1 may prove to be of prognostic value in OSA and could represent an important causal pathway linking the high prevalence of T2D/MetS in OSA. 

Despite an abundance of epidemiological and mechanistic research indicating an independent relationship between OSA and hyperglycaemia/T2D progression, inconsistent data from CPAP intervention trials undermine this association [40]. As CPAP mitigates two of the primary pathogenic mechanisms in OSA, i.e., IH and SF, it would appear conceivable that treatment would positively modulate glucose metabolism [40]. However, CPAP has not been shown to consistently improve glycaemic control or decrease IR in longer-term trials [82,83,84]. A sub-study of the multi-centre RCT (SAVE) with 2687 participants and median of 4.3-year follow-up is the largest and longest study to date exploring the long-term effect of CPAP on glycaemic control and T2D risk. Herein, subjects with OSA and established CVD were stratified to CPAP therapy plus standard care or standard care only. At follow-up, with median CPAP adherence of 3.5 ± 2.3 h/night, treatment failed to improve glycaemic parameters among subjects with OSA, CVD and T2D or pre-diabetes and did not curtail progression from pre-diabetes to T2D compared with standard care [83]. It is important to note, however, that patients exhibiting severe nocturnal hypoxemia and/or excessive daytime sleepiness were excluded, which may limit the generalisability of the findings, as the enrolled cohort may not be entirely representative of patients presenting in clinical practice [82]. Additionally, other studies characterising OSA severity by nocturnal hypoxemic burden have associated IR and T2D only among sleepy subjects [83]. Likewise, dietary habits and physical activity levels were unaccounted for, which is pertinent, as habitual daily activity is associated with insulin sensitivity in pre-diabetes [84,85]. Lastly, CPAP adherence, even within the optimised setting of a trial, was poor, further limiting the findings. Thus, it is difficult to conclude whether the lack of improvement transpires because OSA does not significantly alter glucose regulatory pathways or whether the suboptimal adherence to CPAP therapy and lack of adjustment for possible confounders attenuates potential metabolic benefits [82,83]. More recently and largely in keeping with these data, a 2021 cross-sectional analysis of the RISE clinical trial subjects with pre-diabetes or treatment-naive T2D established no relationship between OSA severity or sleep duration with insulin sensitivity, pancreatic β-cell dysfunction or alternate markers of glycemia, despite the use of gold-standard techniques to assess insulin secretion and β-cell function [84]. Obesity was highly prevalent in this cohort; thus, these findings may not pertain to more healthy-weight OSA patients. As such, it is possible that there is a “ceiling effect” of obesity whereby once certain levels of excess adiposity amass, the impact of OSA or reduced sleep on IR/β-cell dysfunction becomes less significant [84].

With a globally increasing burden of pre-diabetes and T2D, it is imperative to carry out well-designed studies investigating the impact of OSA on glucose metabolism and the role of various treatment modalities such as CPAP or commonly prescribed anti-diabetic drugs such as metformin in slowing the progression of these conditions. Additionally, identifying underlying pathogenic mechanisms could provide novel targets on which to act in the future and aid in the recognition of the patients most susceptible to intervention.

## 4. Leptin

In recent years, a growing body of evidence has implicated the satiety hormone, leptin, in OSA pathogenesis [86,87]. Leptin is an anorexigenic hormone synthesised and secreted chiefly by white adipose tissue and expressed in lesser amounts by additional tissues, including the pancreas, placenta and gastric mucosa [88]. Circulating plasma leptin levels are generally proportional to body fat mass, though accumulating research shows that OSA-associated IH plays a role in leptin upregulation [88]. Data from animal models demonstrate that acute and chronic exposure to IH alters EB, increasing plasma leptin and activating leptin signalling pathways [86]. While obesity is a well-established risk factor for OSA development, these findings infer that a reciprocal relationship exists. Several authors have speculated that altered serum leptin levels and/or leptin-receptor sensitivity secondary to recurrent bouts of IH could predispose OSA-affected individuals to weight gain and metabolic dysfunction irrespective of initial body fat mass [86,89]. In the periphery, leptin modulates CB chemosensitivity, in turn stimulating the SNS, in addition to acting centrally on the hypothalamus and brainstem to regulate energy metabolism [90]. Furthermore, leptin acts a potent respiratory stimulant, modulating ventilatory drive and upper airway resistance and patency via central respiratory nuclei and the peripheral nervous system [90]. Leptin-deficient mice (*ob*/*ob*) exhibited a propensity to respiratory disorders along with reduced ventilatory response to hypercapnia, and leptin administration in these ob/ob models improved the aforementioned respiratory aberrations [21]. In keeping with this concept, lipodystrophic individuals presented with chronically decreased leptin levels and an increased risk of OSA, implying a role of impaired leptin function in OSA aetiology [21,28]. However, the regional deposition of neck fat and IR in these individuals hindered the determination of leptin’s individual significance in OSA development in this cohort [21,28]. The majority of human studies report serum leptin levels up to 50% higher in OSA patients than controls; however, as leptin exhibits diurnal variation, these findings are often confounded by study timing and conditions surrounding leptin measurement [90,91]. Still, increased serum leptin appears to correlate with various indices of OSA severity, including the AHI and oxygen desaturation index [90,92]. However, whether hyperleptinemia in OSA simply reflects the presence of excess adipose, which is often observed in OSA patients, or is truly consequent to IH and SF remains a matter of debate [86]. 

Reports of elevated leptin levels in non-obese OSA patients support the latter supposition, though the regional deposition of fat in these non-obese OSA patients may partly explain these findings [93]. The observed increases in leptin are not reflective of the heightened central metabolic and respiratory effects in OSA patients, however, but rather point to attenuated leptin signalling (leptin resistance) or a “functional deficiency”, in which the protective roles of the molecule are abrogated via downregulation of leptin cellular responses [88]. Accordingly, this relative leptin-deficient or -resistant state is thought to contribute to the genesis of obesity hypoventilation syndrome, whereby ineffective leptin axis functioning within the CNS is thought to provoke hypoventilation [94]. The deleterious effects of OSA–hyperleptinemia are not confined to the respiratory system either. In fact, contemporary research positions leptin as a predictive biomarker for MetS, closely correlated with various MetS sequalae, including IR [95]. Additionally, leptin resistance is thought to contribute to worsening lipid profiles in OSA, with a recent study highlighting that leptin levels in OSA positively correlate with triglyceride levels and negatively correlate with HDL-C [87]. Several intervention trials measuring circulating leptin levels after CPAP therapy have yielded conflicting results. 

Although some trials have reported declining leptin levels within days of CPAP initiation, which persisted with long-term CPAP use, other studies have failed to identify treatment benefits after correction for adiposity [89,96]. Moreover, though CPAP remains the primary treatment modality for OSA patients, long-term compliance amongst users is highly variable, limiting its clinical utility in practice. In such cases, patients may opt for various reconstructive surgical procedures, including uvulopalatopharyngoplasty and genioglossus muscle advancement, which have shown promising results in both alleviating breathing disturbance during sleep and improving the cardiometabolic risk profiles of patients. In a 2019 retrospective chart analysis of 80 patients who underwent OSA surgery, 40 patients categorised as surgical responders (defined as decrease in the AHI of >50% from the pre-operative level and an AHI of less than 20/h in a follow-up sleep study at least 3 months post-operatively) demonstrated significantly improved metabolic profiles 3 months post-operatively, including reductions in LDL-C, triglycerides and leptin, without significant changes in BMI [97]. Consistent with these data, there are similar reports from a prospective study conducted in twenty-three patients undergoing genioglossus muscle advancement with anterolateral advancement pharyngoplasty for OSA. At the 3-month follow-up, serum leptin levels decreased independently of BMI. Additionally, the level of leptin reduction correlated with post-operative airway obstruction improvement in terms of AHI and minimum SaO_2_ [98]. Taken together, these studies indicate that leptin could be useful biomarker of surgical treatment efficacy in OSA, in addition to the traditionally used metrics of polysomnography parameters. However, the data available to date are scant and confined to single-centre studies, with limited follow-up (maximum 3 months) and small-sample-size populations. Given the increasing recognition and implications of the pathophysiological consequences of hyperleptinemia, further research is warranted to elucidate the role of leptin in OSA pathogenesis, leptin-mediated comorbidities in OSA and its potential value as a biomarker of metabolic risk and treatment efficacy.

## 5. Hypertension

OSA exerts pathophysiological effects on the cardiovascular system through an interplay of mechanisms [99,100]. Considerable evidence implicates OSA as an important secondary cause of hypertension (HTN) [101]. These two conditions frequently co-exist: an estimated 50% of OSA patients have hypertension and 30% of hypertensive patients have OSA [101]. Furthermore, hypertensive patients with co-morbid OSA are at increased risk of developing resistant hypertension, diagnosed when HTN persists despite concurrent use of three anti-hypertensive drug classes [101,102]. The OSA-HTN link has been contested by some authors given that its based on cross-sectional studies which cannot infer causality and because CPAP evokes mostly modest reductions in BP in intervention trials. This is in addition to the complexity of disentangling what proportion of BP increase can be definitively ascribed to OSA, obesity or the cross-talk between these entities in studies [100,103,104,105]. Still, data from the large prospective Wisconsin Sleep Cohort (WSC) study provide relatively robust evidence of a temporal relationship between OSA severity and blood pressure (BP). At the 4-year follow-up, the odds ratios (ORs) for incident hypertension among patients with mild–severe OSA were 2.89 (95% CI, 1.46–5.64) and 1.42 (95% CI, 1.13–1.78) in patients without sleep-disordered breathing, irrespective of BMI, WC, NC, age, sex, baseline hypertensive status and other confounders [106]. Reaffirming these findings, a 2018 meta-analysis of 26 studies and 51,623 participants presented a significant association of OSA with essential hypertension with pooled ORs of 1.184 (95% CI, 1.093–1.274) for mild OSA, 1.316 (95% CI, 1.197–1.433) for moderate OSA and 1.561 (95% CI, 1.287–1.835) for severe OSA [107]. In the SHHS, increased odds of HTN were reported with the increase in OSA severity; however, the risk dissipated and became non-significant after adjustment for BMI [104]. The incongruence between the studies may be attributable to the older average age of participants in the SHHS (60 SHHS vs. 47 WSC), which in another study also attenuated the strength of the association between OSA severity and HTN [106,108,109,110]. Furthermore, the SHHS consisted of a multi-ethnic cohort when compared with the homogenous WSC study population, which could further contribute to the variance. 

The mechanisms by which OSA may contribute to HTN are multifactorial. Excessive sympathetic excitation appears to be the most noteworthy, with further potential involvement of the renin–angiotensin–aldosterone system (RAAS), and altered gut microbiome and inflammation [111]. In healthy individuals during sleep, there is a decrease in sympathetic activity and parasympathetic predominance, which contributes to the physiologic nocturnal “dipping” in BP and heart rate (HR). The absence or abating of this night-time dipping pattern is common in OSA and has been shown to confer CVD risk [112]. Periodic intermittent hypoxia (IH) and hypercapnia due to apnoea–hypopnea episodes elicit autonomic alterations which antagonise the natural dipping phenomenon [113]. Chemoreflex-mediated sympathetic nerve activation (SNA) secondary to IH and subsequent serum catecholamine elevations acutely increase cardiac output and peripheral vascular resistance, contributing to increased BP. High serum and urinary catecholamines in OSA persist even beyond the nocturnal period to wakefulness and positively correlate with OSA severity [114]. This capacity of catecholamines to outlast the initial hypoxic stimulus fosters the transition from night-time HTN to overt HTN in OSA [114]. In line with this, the presence of hypertension in such patients may signal a predisposition to more severe cardiovascular consequences, given the commonly reported “non-dipping” hypertension pattern—defined by a nocturnal blood pressure reduction of less than 10%—observed in those with OSA [112]. Chronic and repetitive surges in BP resulting from heightened SNA may also impair the function of arterial baroreceptors, which normally act to detect variations in carotid or aortic stretch prompted by increases or decreases in arterial pressure [112]. Moreover, UA occlusion initiates an abrupt increase in negative intra-thoracic pressure (−60–80 mmhg), which alters cardiac afterload. Mechanically taxing events such as acute atrial and ventricular wall stress and myocardial O_2_ demand increase, repeated over time, can precipitate left ventricular hypertrophy and cardiac remodelling [100]. In addition, it has been documented that patients with OSA have increased plasma renin levels mediated by chronic IH-induced sympathetic outflow to the kidney. Consequently, angiotensin-II and aldosterone levels increase. Combined, these may contribute to HTN by means of vasoconstriction and fluid retention, respectively [101]. In turn, aldosterone may also worsen pre-existing OSA by increasing parapharyngeal oedema and promoting further UA occlusion [115]. Medications which modulate RAAS pathways, particularly mineralocorticoid antagonists, spironolactone and eplerenone, have produced some encouraging results in lowering BP and the frequency of apnoeic events in OSA-HTN patients, albeit within very-limited-sample-size trials with a maximum of a 3-month follow-up [116,117,118]. OSA-induced hypoxia also affects the microvasculature by causing inflammation and oxidative stress, subsequently leading to decreased NO and increased endothelin-1 production through extensively discussed molecular mechanisms [119].

Intriguingly, in recent years, accumulating evidence suggests that OSA-induced inflammation and gut dysbiosis may play a role in the development and progression of HTN in OSA. Though much of this research remains in its infancy, Ayyaswamy et al. demonstrated that two weeks of OSA in a murine model resulted in perturbations in gut microbiota, promoting a pro-inflammatory environment and HTN development [120]. In the same study, after identification of elevated TH17 cells in both the gut and brain, neutralisation of IL-17a with antibodies decreased gut inflammation and neuroinflammation, in addition to preventing HTN development, suggesting a central aetiological role for IL-17a in OSA-induced HTN [120]. Moreover, various studies have demonstrated increased intestinal permeability, microbial translocation and *Firmicutes*-to-*Bacteroidetes* ratio alongside blunted gut microbial diversity in OSA hypertensive populations [121]. Given the ramifications of these gut alterations in OSA and the potential contribution to HTN, future research should expand our understanding of this complex interplay and ascertain if restoration of the microbiota through probiotic supplementation or fibre-rich diets could attenuate the future risk of HTN in OSA patients. However, it is noteworthy that research on gut microbiome is burdened by multiple technical challenges in defining the gut microbiome composition. Specifically, Walters et al. demonstrated that the per-study effect exceeds the biological effect in terms of gut microbiota composition in obese individuals [122,123]. On top of that, additional bias may emerge due to diverse variations in data analysis and sample processing, including differences in DNA extraction methods, sequencing techniques, primer selection and bioinformatic analysis [124]. The complex interface of putative associations between OSA and components of MetS is summarised in Figure 2.

## 6. Dietary and Lifestyle Modification in OSA and MetS

The occurrence of OSA and MetS is strongly associated with obesity, and excess body weight represents the strongest modifiable predictor of both ailments [12]. As discussed, obesity imposes mechanical stress on the UA and co-occurring obese physiology results in the genesis of a dysmetabolic state. As such, weight loss attained from either lifestyle intervention (diet and/or exercise) or pharmacological treatments should be considered central to OSA/MetS to mitigate onset, severity and future cardiometabolic risk [125]. Current obesity guidelines advocate 5–10% weight loss over 6 months to provide health benefits for most overweight and obese individuals; however, beyond this consensus, official and specific dietary guidelines for MetS and OSA are lacking [126]. Several studies have established marked improvement in OSA parameters in weight loss intervention trials, in addition to improved MetS components and metabolic-associated fatty liver disease (MAFLD) [127]. Early clinical trials predominantly evaluated the effects of weight loss on OSA severity induced by significant energy restriction in the form of very-low-calorie diets (VLCDs) [127]. Though these trials have exhibited clinically meaningful reductions in OSA severity, a key concern with implementing intense caloric restriction is failure of long-term weight maintenance after discontinuation [128]. However, in a 2015 trial, Fernandes et al. demonstrated that even moderate energy restriction can significantly reduce the AHI and increase minimum O_2_ saturation with less overall weight loss compared with prior VLCD trials [127]. Adding further credence to these findings, Dixon et al. reported that laparoscopic adjustable gastric banding, when compared with conventional weight loss therapy, did not produce statistically greater reductions in the AHI, despite 6-fold greater weight loss between treatment groups (27.8 kg in bariatric surgery group vs. 5.1 kg conventional weight loss program) [129].

Thus, additional factors other than absolute weight loss, including the site of adipose reduction (central vs. peripheral), may influence the degree of OSA improvement [127]. These findings are in contrast with evidence emanating from secondary analyses of the Management of Obstructive Sleep Apnoea (MIMOSA) RCT, which evaluated the effectiveness of a weight loss/lifestyle intervention (Mediterranean diet) plus CPAP compared with CPAP alone for OSA management. The trial revealed a dose–response relationship between improvements in OSA symptoms and severity and the degree of weight loss, with the greatest benefit being achieved by patients with ≥10% weight loss [130]. Similarly, high-sensitivity CRP levels and urinary 8-iso prostaglandin F2a levels were lower among participants randomised to the Mediterranean lifestyle intervention group (Mediterranean diet, counselling on physical activity and sleep habits), highlighting additional anti-inflammatory and antioxidant benefits of the Mediterranean diet in OSA, independently of weight loss [131]. Perhaps most importantly, investigation of cardiometabolic parameters 6 months post-intervention revealed sustained improvements in both intervention arms of the study (Mediterranean diet group (MDG) and Mediterranean lifestyle group (MLG)) with lower insulin, triglycerides and Hs-CRP and higher HDL noted, in addition to reduced relative risk of MetS, regardless of some weight regaining among participants [132]. Thus, while CPAP prescription remains the standard of care, clinicians should encourage weight loss, ideally ≥10%, to ameliorate OSA symptom and severity. Moreover, though the optimal diet is the considered the diet that a patient can adhere to, added cardiometabolic benefits and MetS risk reduction may be derived from adopting a Mediterranean-style dietary pattern, which emphasises inclusion of plant-based foods and oils, whole grains, fruits, vegetables, nuts and seeds. 

## 7. Future Directions and Conclusions

Although multidirectional associations exist between OSA and components of metabolic syndrome, the exact mechanisms and, more importantly, the relative contribution of each to the pathophysiology of OSA and its consequences are still unknown. Despite the fact that the use of CPAP not only mitigates the AHI but also has some favourable metabolic effects, CPAP adherence is very low (<20%), and its use has limited efficacy in treatment of OSA-associated MetS. Hence, a further understanding of the previously discussed mechanisms is warranted in order to improve quality of life and outcomes in these patients. Alternative approaches to OSA treatment have become highly appealing with the advent of glucagon-like peptide-1 receptor agonists (GLP-1 RAs), incretin-based drugs that have revolutionised obesity management. Weight loss represents the primary mechanism through which GLP-1 RAs may alleviate OSA. Furthermore, previous research suggests that GLP-1 RAs could ameliorate OSA through various weight-independent mechanisms, most notably improved insulin sensitivity, suppression of increased oxidative stress and GLP-1RA-mediated regulation of sleep and wakefulness [133,134,135,136]. Finally, given their pleiotropic cardioprotective effects, it is plausible that the use of GLP-1 RAs in these patients could enhance cardiovascular outcomes independently of other treatment options [137,138,139]. The potential drawbacks of GLP-1 RA use are gastrointestinal adverse effects and the fact that a significant number of patients regains most weight after discontinuing the medication [140,141]. As there are limited data on the impact of GLP-1 RAs on OSA, ongoing trials are currently investigating the potential of GLP-1 RAs in these patients. For instance, the SURMOUNT-OSA (NCT05412004) study, a 52-week, phase III randomised controlled trial, is underway to explore the efficacy and safety of tirzepatide (a GLP-2 and GIP receptor agonist) in the treatment of OSA. Similarly, sodium-glucose co-transporter-2 inhibitors (SGLT2is), a group of medications that has been shown to offer major cardio and renoprotective effects in recent trials, have also demonstrated encouraging results in OSA [142,143,144]. Putative mechanisms include weight reduction, reduced systemic fluid retention, nocturia-mediated suppression for the duration of REM sleep, suppression of circadian SNS activation, reduced CO_2_ production and modulation of arousal threshold [145,146,147,148]. Overall, data with sufficient power to detect the risk–benefit profile of SGLT2is and GLP-1 RAs remain to be elucidated. While awaiting the results of ongoing RCTs, real-world observational studies analysing the existing databases can provide valuable information in this setting.

Apart from weight loss-targeting drugs for OSA, combinations of noradrenergic and anti-muscarinic agents have emerged in recent years as potential pharmacotherapeutic options to ameliorate OSA severity [149]. During sleep, the withdrawal of endogenous norepinephrine on genioglossus muscle tone, alongside the inhibitory action of muscarinic receptors, results in decreased pharyngeal and tongue muscle tone, which provided the mechanistic rationale for the use of these drug regimens. A 2023 meta-analysis of eight studies depicted a favourable response to various combinations, with reductions in the AHI, hypoxic burden and lowest O_2_ saturation. However, seven of the included studies were 1-night interventions, while the remaining one evaluated the effects over the course of 1 week; thus, it is unclear whether the observed effects decline with time. In the future, longer-duration and larger trials with specific combinations of these drug classes should address such questions in addition to potential adverse effects related to sustained use of anti-muscarinic and noradrenergic agents in OSA patients [150]. 

Additionally, research to date has identified several anatomical and physiological features contributing to the heightened prevalence of OSA in men, including the tendency for a more central pattern of fat distribution, a lack of protective progesterone and estradiol compared with pre-menopausal women and higher Pcrit in males [151,152]. With that said, the gender gap dwindles significantly after menopause, when prevalence of OSA in women doubles [152]; yet, post-menopausal women remain under-represented in the majority of OSA research. Moreover, evidence suggests that female individuals with OSA are often underdiagnosed and treated secondarily to more “atypical” presentations, though they are not less symptomatic [152]. Finally, female individuals with OSA in a recent cross-sectional analysis were more likely to have MetS (88.27%) compared with male OSA patients (66.38%) [153]. Enhancing our understanding of the existing gender differences in OSA and its consequences will aid in recognition, intervention, and more targeted personalised therapies for OSA patients. 

In summary, there is an inextricable bidirectional connection between OSA and components of metabolic syndrome. Obesity seems to promote OSA development via various pathways, some of which are not even directly related to mass but rather to metabolic complications of obesity. Simultaneously, OSA promotes weight gain through central mechanisms, forming a potentially vicious cycle. On the other hand, diabetes contributes to OSA pathophysiology mainly through effects on peripheral nerves and carotid body desensitization, while IH and SF are the principal culprits in OSA-mediated T2D. Importantly, apart from this bidirectional pathophysiological relationship, both obesity and T2D together additively increase cardiovascular risk in this population. The emergence of new drugs targeting obesity and unequivocal results of the available studies underscore the need for further exploration of the mechanisms linking metabolic syndrome and OSA, all with the aim of improving outcomes in these patients.

## Figures and Tables

**Figure 1 ijms-25-03243-f001:**
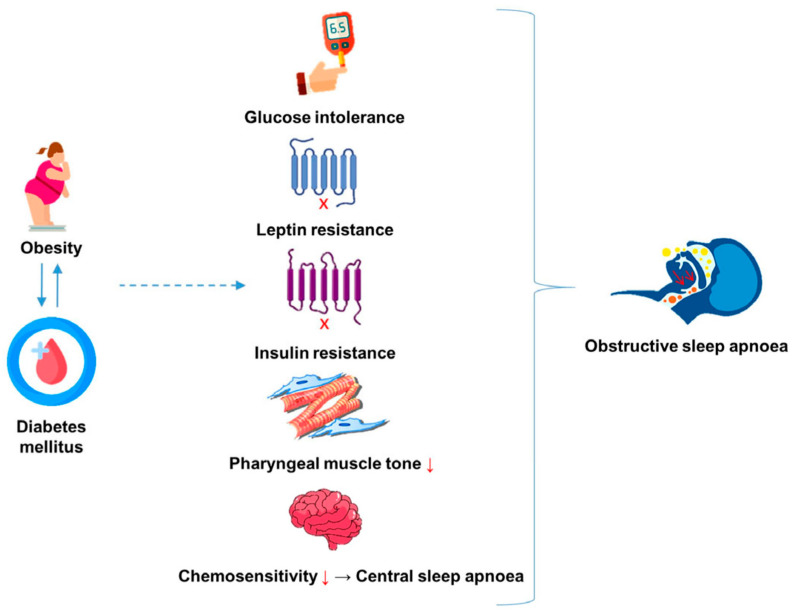
“Weight-independent” mechanisms by which components of metabolic syndrome contribute to development of obstructive sleep apnoea. “X” represents impaired signalling through the respective receptor whereas arrows indicate a decline.

**Figure 2 ijms-25-03243-f002:**
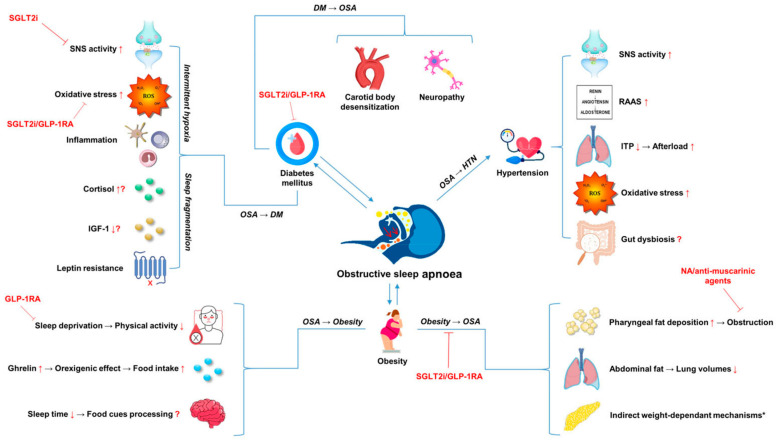
The complex interface of putative associations between OSA and components of metabolic syndrome. * See Figure 1. Abbreviations: DM: diabetes mellitus; GLP-1RA: glucagon-like peptide-1 receptor agonist; HTN: hypertension; IGF-1: insulin-like growth factor 1; ITP: intra-thoracic pressure; NA: noradrenergic; OSA: obstructive sleep apnoea; RAAS: renin–angiotensin–aldosterone system; SGLT2i: sodium-glucose co-transporter-2 inhibitor; SNS: sympathetic nervous system.

## Data Availability

Not applicable.

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
