# Peer review of "Metabolic Crossroads: Unveiling the Complex Interactions between Obstructive Sleep Apnoea and Metabolic Syndrome"

_ijms, 2024, doi:10.3390/ijms25063243_

Round 1

Reviewer 1 Report

Comments and Suggestions for Authors

In this manuscripts, Heffernan and colleagues discuss the complex interactions between obstructive sleep apnea and metabolic syndrome. They conclude that the emergence of new  drugs targeting obesity and unequivocal results of the available studies underscore the need for further exploration of the mechanisms linking metabolic syndrome and OSA, all with the aim of improving outcomes for these patients. The manuscript is interesting. It is well-written and well-organized. I want only to rise minor points that might improve the quality of the manuscript.

-       The Authors should better discuss the state of the art of pharmacological treatment of OSA (DOI: 10.1016/j.smrv.2023.101809) and metabolic syndrome.

-       The Authors must better discuss the positive impact of diet on metabolic syndrome, OSA and  sleep disorders (DOI: 10.2147/NSS.S325494; DOI: 10.3233/JBR-220054 and others).

-       The Authors should also better discuss the role of sex in the interplay between OSA and metabolic syndrome. It is imperative to study sex differences in order to develop personalized treatments.

-       Please check for typos throughout the manuscript.

Comments on the Quality of English Language

Minor editing

Author Response

Reviewer 1

We thank the Reviewer for providing a valuable insight to our manuscript. We hope that after these changes the paper will be suitable for publication.

The Authors should better discuss the state of the art of pharmacological treatment of OSA (DOI: 10.1016/j.smrv.2023.101809) and metabolic syndrome.

  • We thank Reviewer 1 for this valuable suggestion about recent advances in pharmacotherapy for OSA and MetS, and we have now added an additional paragraph in the future directions section to discuss noradrenergic and anti-muscarinic drug regimens [see line 585-597]

The Authors must better discuss the positive impact of diet on metabolic syndrome, OSA and sleep disorders (DOI: 10.2147/NSS.S325494; DOI: 10.3233/JBR-220054 and others).

  • In accordance with Reviewer 1s comments concerning the lack of discussion regarding the positive effects of  dietary modifications in OSA and MetS an additional section has been included to address this issue [see section 6, lines 505-551]

 The Authors should also better discuss the role of sex in the interplay between OSA and metabolic syndrome. It is imperative to study sex differences in order to develop personalized treatments.

  • We have added further detail to the previously briefly mentioned sex dimorphism in OSA. A more thorough discussion has now been included in the Future directions and conclusion section, highlighting the under-representation of females in OSA research to date and potential implications of this.

Reviewer 2 Report

Comments and Suggestions for Authors

The authors discuss the complex interactions between metabolic syndrome and obstructive sleep apnea. this is a narrative review in which it would be worth providing information on how the articles were selected.

After reading the article, I can say that it is very well written and its content is understandable. Additionally, the authors have included diagrams that make it easier to view the content.

My comments are quite concise and concern expanding the content of the review to include several additional issues

1B. I missed the reference to metformin used to treat diabetes

1A. perhaps statins used to treat atherosclerosis should also be discussed

2. I would add a diagram or a table to the drugs discussed

3. and a discussion of fatty liver disease that accompanies the metabolic syndrome should be added

Author Response

Reviewer 2

We thank the Reviewer for providing a valuable insight to our manuscript. We hope that after these changes the paper will be suitable for publication.

1A. perhaps statins used to treat atherosclerosis should also be discussed

  • We thank Reviewer 2 for this suggestion, however the most recent evidence in OSA pathophysiology and pharmacotherapy (GLP-1 agonists, SGLT-2 inhibitors, noradrenergic and anti-muscarinics) in addition to mineralocorticoid antagonists for HTN in OSA have been discussed throughout our manuscript. As such, we feel an additional discussion of statins for atherosclerosis outside the remit of our current review.

1B. I missed the reference to metformin used to treat diabetes

  • While newer antidiabetic medications were discussed in the Future directions & conclusion section of the manuscript however as Reviewer 2 has acknowledged we failed to mention Metformin throughout the text. We have now described this as a commonly used anti-diabetic medication in Section 3. OSA Hyperglycemia and Diabetes Mellitus (Lines 330-333).

2. I would add a diagram or a table to the drugs discussed

  • Dear Reviewer, thank you for your valuable insight, we have now expanded Figure 2 with appropriate mention of therapeutic agents aimed at OSA treatment.

3. and a discussion of fatty liver disease that accompanies the metabolic syndrome should be added

  • We would like to thank Reviewer 2 for this suggestion however we feel that a discussion of fatty liver disease in addition to Metabolic syndrome and OSA is beyond the scope of this review which focuses primarily on the bidirectional relationship between two conditions, OSA and Metabolic Syndrome. We would like to direct the reviewer to other papers such as Chung et al., 2021 (https://www.nature.com/articles/s41598-021-92703-0) which explore the relationship between NAFLD/ MAFLD and OSA and cover this topic in much greater detail.